# Thinking about life in COVID-19: An exploratory study on the influence of temporal framing on streams-of-consciousness

Constance M. Bainbridge[ID]*, Rick Dale

Department of Communication, University of California, Los Angeles, Los Angeles, California, United States of America

* cbainbridge@ucla.edu

## Abstract

The COVID-19 global pandemic led to major upheavals in daily life. As a result, mental health has been negatively impacted for many, including college students who have faced increased stress, depression, anxiety, and social isolation. How we think about the future and adjust to such changes may be partly mediated by how we situate our experiences in relation to the pandemic. To test this idea, we investigate how temporal framing influences the way participants think about COVID life. In an exploratory study, we investigate the influence of thinking of life before versus during the pandemic on subsequent thoughts about post-pandemic life. Participants wrote about their lives in a stream-of-consciousness style paradigm, and the linguistic features of their thoughts are extracted using Linguistic Inquiry and Word Count (LIWC). Initial results suggest principal components of LIWC features can distinguish the two temporal framings just from the content of their post-pandemic-oriented texts alone. We end by discussing theoretical implications for our understanding of personal experience and self-generated narrative. We also discuss other aspects of the present data that may be useful for investigating these thought processes in the future, including document-level features, typing dynamics, and individual difference measures.

## Introduction

What is the semantic structure of free-flowing thought? How do meanings come up in our thoughts, and how are they linked over time? Human experience is filled with this structure, while we stand in line for our groceries, wait in line at the bus station or even in a moment of mind wandering while conversing with a friend. In this paper, we explore recent events surrounding COVID as a domain to tap into this semantic structure. We devised a "stream-of-consciousness" task in which participants imagined the future beyond COVID, and quantified how their free-flowing thought varied as a function of how they were prompted before this writing. We find that if participants are prompted with the present COVID situation vs. the pre-COVID times, their structure of thought changes. Our exploratory analyses suggest

**Data Availability Statement:** The data and analysis script is available on Github: https://github.com/conbainbridge/covid_thoughts DOI: 10.5281/zenodo.7809317.

**Funding:** The authors received no specific funding for this work.

**Competing interests:** The authors have declared that no competing interests exist.

language from the future-oriented responses reflects its temporal priming, i.e., the pre-pandemic vs. during the pandemic prompt that came before. These results offer hints at the semantic patterns that characterize these self-reflections, and how context is central to the forms they take. We end by arguing that a generalized notion of "self-communication" may organize phenomena such as these in intriguing ways.

## Background

When the COVID-19 global pandemic spread rapidly in late 2019 and early 2020, major life impacts reverberated globally. These effects were felt across many aspects of everyday life, from direct health impacts to more indirect effects on the economy and social life. People began referring to life before the pandemic colloquially as the "before times," and there was a sense of a new normal. These effects were also significant in the lives of younger individuals, such as students, with virtual schooling, diminished social interaction, and limited hands-on learning (e.g., [1]). Evidence suggests an alarming impact of the pandemic on the mental health of college students, including increased stress moderated by self-regulation efficacy [2], increased depression and anxiety [3], and negative changes to student relationships [4]. The effect of perceived threat of COVID-19 on mental well-being appears to be mediated by future anxiety as well, showing the potential to impact mental health in the long run as decisions about the future may be impacted [5]. The pandemic thus provides a unique opportunity to study how major events influence perceptions of life and mental health, and their relationship to other dimensions.

To study these perceptions, we investigate how student participants construct a narrative text about their lives. Our approach is inspired by methods used in essay writing and journaling [6], self-talk [7], and think aloud [8]. These domains suggest that when we speak to ourselves, ruminate, or reflect on aspects of our lives, our linguistic styles and strategies may be a signature for underlying mental or emotional states and processes. Intriguingly, such intrapersonal communication has been frequently the topic of discussion, yet remains largely an understudied construct. Self-communication occurs when both sender and receiver of a communicative instance are contained within a single individual, such as in dialogical self-talk [7], and can include transcending across time and space [9]. Here we use this process as a source of data about these life perceptions.

While not all aspects of intrapersonal communication may be easily accessible for study, such as the seemingly endless streams-of-consciousness we engage in every day, various methods have been employed to tap into self-talk through writing or speech. Raffaelli et al. [8] used a think aloud paradigm in which individuals are instructed to speak aloud their thoughts. Negative valence in the words used correlated with a narrowing of conceptual scope, such that thoughts became more semantically similar when they were more negative. Social context may play a role, too. Oliver et al. [10] used a think aloud task to study mental health outcomes. Changing social context to more supportive environments led to greater use of positive emotion words, fewer negative emotion words, fewer swear words, and fewer first-person references compared to a control condition that lacked emotional recognition or meaningful rationale. Recent work has shown that self-talk also links to mental health outcomes during COVID-19. In a questionnaire study conducted on an Iranian sample, there were significant relationships found between self-talk, death anxiety, obsessive-compulsive disorder, and coping strategies in relation to the pandemic [11].

As noted above, the onset of COVID for some may present a distinct point in time at which everyday life changed. This temporal effect of COVID may alter the way we contextualize and think about events before, during, and after this distinct transition. Changing the temporal

framing of one's thoughts or self-talk may have an influence on the language that we use. For example, construal-level theory hypothesizes that increased psychological distances are linked to increased abstractness of hypotheticals [12]. The further away something is in time, space, or relatability (e.g., feeling similar to or different from an individual), the further the perceived psychological distance and the less concrete related thoughts become. Similarly, increasing concreteness, such as through writing about a given event, may decrease psychological distance to that event in time [13]. High-level construals, or higher abstractions when mentally representing objects or concepts, may better facilitate self-control, such as attenuating the impact of future discounting on economic decision making [14]. Following the September 11, 2001 attacks on the United States, entries in an online journaling platform used increasingly psychologically distant language in their daily writings, suggesting major events can impact our relationships with time [15]. However, prior work suggests that writing about emotional experiences such as trauma may provide benefits for both mental and physical health [16].

Considering these findings, one might expect that writing about "the before times," pre-pandemic, would influence how one perceives their future, perhaps with increased abstraction leading to greater possibilities and allowing distance from such a troubling and disruptive event. Relatedly, a focus on life during the pandemic may have negative impacts on one's perceptions of their future after the pandemic, with lower levels of construal leading to ruminative tendencies.

To test this idea, we collect and analyze a text-based "stream-of-consciousness" dataset. Participants carried out this open-ended response task online, typing in their thoughts about COVID-19 life under different temporal conditions. The task was designed to elicit a naturalistic and uninterrupted flow of thought. Participants were first told to consider and write about life either before or during the pandemic. This prompt (before vs. during) served as a frame for a subsequent writing prompt, where participants were instructed to share their thoughts about a post-pandemic life. This future-oriented prompt is the same for all participants and is the focus of our analysis, and participants only differed in which writing prompt preceded this one (before vs. during the pandemic). This open-ended writing task generates a large and rich dataset of text. We thus took a preliminary, exploratory approach to investigate the influence of this temporal framing on their responses. In the *Analyses* section, we consider prior research that frames some factors guiding our exploratory analysis, and we introduce the ways in which these texts can be measured and analyzed.

## Methods

Data collection was conducted during two separate college quarters: in the first quarter (fall, 2021), classes were hybrid (both in-person and online), with students returning to campus for the first time since the pandemic began. This research was approved by the UCLA North General Institutional Review Board. 134 undergraduate students (female = 95, male = 38, other = 0) contributed data to this first phase of sampling. The second phase of sampling occurred the following quarter (winter, 2022), which had returned to online-only for the first four weeks due to the rapid spread of a particularly contagious strain of the virus, labeled "omicron." In this phase, 91 undergraduate students (female = 70, male = 19, other = 2) contributed data. The students completed the study online for course credit in an introductory communication course. The goal of the study was to collect a rich dataset for exploratory analyses, and several aspects of the data were not included for analysis. Because of the pandemic-related constraints participants encountered at the start of the winter quarter, we first use this second phase dataset for our main analyses. We then use the fall dataset for exploratory comparison.

The experiment was built using jsPsych [17] in conjunction with https://cognition.run to store the data. First, participants encountered a consent page, then click to continue only if they agree to consent to participate in the study. After the initial consent page, participants selected on a slider where in the COVID-19 timeline they considered the current moment to be. For the main portion of the study, participants wrote in a stream-of-consciousness style manner for ten minutes per prompt, responding to three total writing prompts. The first prompt asked participants to write either as if it is before the pandemic, or as if during the pandemic. Following the initial prompt, participants responded to a similar prompt asking to write as though it is after the pandemic. The final prompt included whichever temporal framing was not responded to in the first prompt. This resulted in two possible conditions: before-after-during the pandemic, or during-after-before the pandemic. Our focus here is on how the before vs. during prompt, chosen randomly as the first temporal framing, influences the way participants write about the future, after the pandemic.

During all writing prompts, a countdown timer was visible on the screen during writing, and on multiple pages throughout the study, mental health resources were provided. After completion of the three prompts, participants responded to questions about demographics, COVID-19 experiences, journaling experience, as well as three individual difference measures: a rumination scale [18], an 18-item adaptation of the need-for-cognition scale [19], and a social connectedness and belonging scale [20].

## Measures and analyses

In the analyses that follow we focus on document-wide features, taking an exploratory approach. Linguistic Inquiry and Word Count (LIWC) [21] categorizes the words in a text based on a range of concepts, including emotions, cognitive tension words, causal words. LIWC provides one methodological tool for enabling indirect inferences about mental states. The most recent version of LIWC at the time of analysis features over 100 word categories, capturing a large variable space. This version of LIWC was tested and validated using a "Test Kitchen Corpus" of around 31 million words pooled from a wide range of corpora, including blogs, emails, movie dialogues, transcribed speech, natural conversations, social media posts, and more [21]. Analysis of language data can be challenging due to the complexities at play. However, LIWC has been successful at predicting a variety of psychological and social measurements from language usage. For example, course performance has been generally predicted based on the written self-introductions of undergraduate students [22].

Several of the specific LIWC word categories also map neatly onto well-studied and meaningful dimensions of language. LIWC includes several sentiment related categories, including positive and negative emotion and tone, as well as several discrete emotions, such as sadness, anger, and anxiety. Sentiment of language may provide hints at wellbeing. In one pair of studies, improvements in physical health were linked to a greater use of positive emotion words and a moderate number of negative emotion words (neither very high nor very low), as well as increased use in both causal and insight words throughout a writing task [23]. Pronoun usage may also hint at different psychological processes. Greater use of first person singular pronouns is associated with interpersonal distress [24], as well as depressive symptoms and negative emotions [25]. LIWC additionally includes categories relating to time, such as a past or future focus, and health categories, all concepts highly relevant for the topics of interest in this dataset. Content words (e.g., nouns, regular verbs, and various adjectives and adverbs) and function words (e.g., pronouns, prepositions, articles, conjunctions, and so on) are also detectable using LIWC, and may reveal information about one's social inclination—the use of function words often requires understanding shared knowledge between interlocutors, for example [26].

For each future-oriented text produced by participants, LIWC generates a set of semantic category measures that reflect the percentage of these categories represented in that text. This can be understood as a multivariate vector of measurements of how positive, negative, etc., a text is based on a calculation of the percentage of words that fall under these categories. To evaluate the influence of temporal framing (i.e., writing about pre-pandemic or during pandemic life in the first prompt) on writing about life after the pandemic, we evaluated the LIWC features present in the post-pandemic documents (the LIWC data for all documents and the analyses script are available at https://github.com/conbainbridge/covid_thoughts). Because LIWC has over 100 of these categories, we face the challenge of multivariate analysis without simply deploying an analysis pipeline on each of the 100 separate dimensions. To conduct a more global analysis of LIWC features, we used principal components analysis (PCA) to determine components that best predict the temporal framing condition. PCA permits the analysis of many intercorrelated variables, characterizing the structure of both the observations in the dataset and the variables themselves (for a detailed explanation, see [27]). PCA has been used successfully in prior work to reduce LIWC dimensions [22, 28, 29].

PCA thus extracts a conceptual space across all LIWC dimensions, but at a lower dimensionality. Another way to think of this process is that PCA reveals this lower dimensionality based on how normalized LIWC scores cluster across students' writing. For example, instead of the three LIWC features "positive tone," "negative tone," and "emotion," the PCA model may infer that these three features load onto just one principal component (PC). This example is intuitive, but finding clusters across texts and many features yields subtler and more complex patterns of correlation. Our main test is whether these lower-dimensional PCs distinguish temporal prompts at all. This analysis is done based solely on the post-pandemic-oriented texts, to see how the framing of a preceding prompt may echo into thoughts about the future. Put simply: It would show that participants primed by the past or present (pre- and during pandemic) alter the way they think (or write about) the future.

## Results

The PCA recovers as many PCs as there are variables, ranked by the strongest component to the weakest. In cases where the number of participants is less than the number of variables, the number of PCs is limited to match this sample size. Because the winter dataset's sample size ($n$ = 91) is less than the total LIWC variables (117), the PCA yields 91 total PCs. The LIWC features cluster across texts as we found a nonlinear rise in PC strength, expressed through cumulative proportion of variance accounted for (Fig 1). The first 20 PCs account for almost 70% of the cumulative variance in the dataset.

As noted above, we tested whether these LIWC PCA components from the future-oriented prompt relate to the temporal frame of the prior prompt (before vs. during the pandemic). With a logistic regression predicting prior condition from these 20 components, we found seven PCs that were significant or approached significance. We chose a liberal initial threshold of $p$ = 0.1 to ensure we captured a wide range of possible semantic structures in the future-oriented writing. In a secondary generalized linear model, six of these remained significant (PCs 1, 4, 5, 10, 11, and 13). We also included the seventh (PC18) in our selected components because it trended towards significance in that follow-up model alone. Note that these results reflect coefficients from a single regression model–not independent tests.

The $p$-values for each PC in the model, and the top ten most influential LIWC features per PC (i.e., highest absolute values in loading scores, ordered from most to least influential) are in Table 1. LIWC categories in bold, italic font feature positive loading scores, indicating their tendency to characterize the *during*-pandemic framing, while negative (normal font) loading

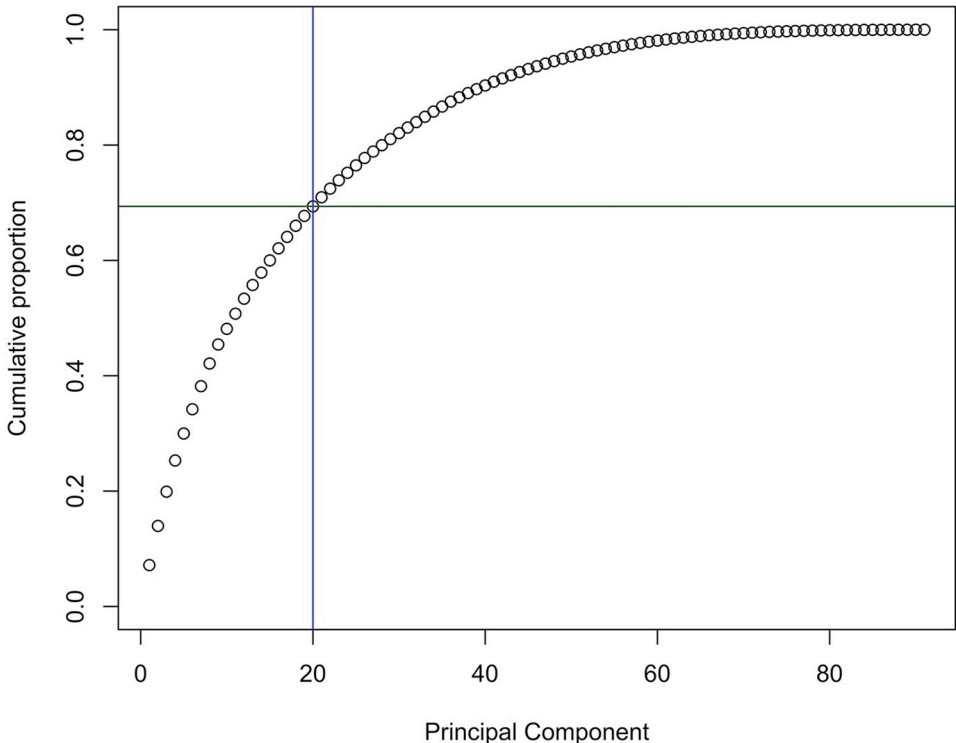

**Fig 1. Cumulative variance of the principal components.** Of the 91 total PCs, a subset of 20 accounting for approximately 69% of the cumulative variance was taken to determine the most significant PCs.

scores characterize the *pre*-pandemic framing. Loading scores are available in the S1 Table. For details on what the different LIWC categories entail, see [21].

PC5 is the most significant component of the selected components from the generalized linear model. By itself, it is able to predict which temporal framing preceded the post-pandemic

**Table 1. The top ten most influential LIWC features for each selected component.** The top ten LIWC features for each PC, ordered from most-to-least influential (i.e., highest absolute loading scores), clustered by the condition they characterize. Features in bold, italic font have positive loading scores and characterize the during-pandemic condition, while the features in normal font have negative loading scores and characterize the pre-pandemic condition. Loading scores for these LIWC features are available in the S1 Table.

|      | P-value       | Top 10 most influential LIWC features |
|------|---------------|----------------------------------------|
| PC1  | * 0.021       | ***Health, Illness, Analytic, Neg. tone, Article, Physical,*** Pronoun, Social refs, Social, Personal Pronoun |
| PC4  | * 0.027       | ***Space, Aux. Verb, Perception, Allure,*** Fulfill, Prosocial, Exclamation, I, Insight, Social behavior |
| PC5  | ** 0.009      | ***Pos. Tone, Want, Tone, Discrepancy, Pos. Emotion, Emotion, Lack,*** Past focus, Authentic, Personal pronoun |
| PC10 | * 0.085       | ***Conjunctions, Acquire, Swear, Risk,*** Communication, Impersonal pronoun, Memory, Sad emotion, Leisure, Lifestyle |
| PC11 | * 0.047       | ***You, Male, Space,*** Achieve, Death, Work, Cognition, Lifestyle, Q mark, Certitude |
| PC13 | * 0.041       | ***Cause, Time, Future focus, Conjunctions, She/he, Need,*** Mental, Visual, Perception, Past focus |
| PC18 | 0.111         | ***Moral, Impersonal pronoun, Certitude, Motion, Words per sentence, Affect,*** Need, You, Personal pronoun, Apostrophes |

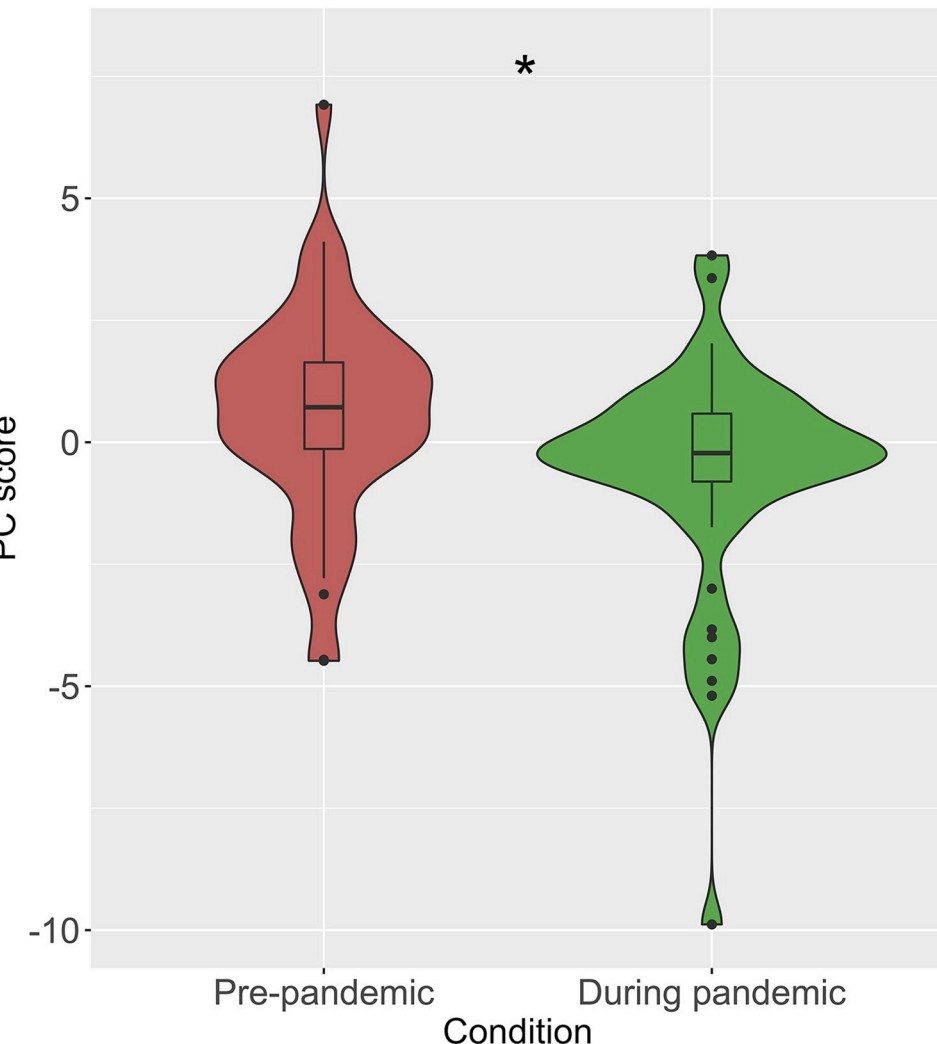

**Fig 2. Comparison of pre-pandemic vs. during pandemic priming of future-oriented writing, as distinguished by PC5.** The temporal framing of during-pandemic represents the reference condition in green, while the pre-pandemic condition is in red. The difference in explanatory power for PC5 between the two conditions is significant at *p* = .011.

prompt, based solely on the linguistic characteristics in that post-pandemic prompt ($p$ = .011, Fig 2). PC5 shows that positive emotion, tone, want, and discrepancy are found more in post-pandemic contemplations when they are preceded by reflections about *during* the pandemic. Conversely, when prompted with *before* the pandemic, PC5 shows focus on the past and use of personal pronouns. Interpreting LIWC loadings may be subjectively influenced, and interpretive assessment must be done with caution. In this particular case, PC5 indicates that thoughts about experience during the pandemic prompt positivity (i.e., the presence of the "positive tone" and "positive emotion" LIWC categories) that is desired ("want", "tone," "emotion," and "discrepancy"–which includes words like would, can, and want). On the other hand, the pre-pandemic priming may focus on what was lacking ("lack") in the past ("past focus") and may be expressing themselves more spontaneously ("authenticity").

An interpretation of PC1 could indicate people think more negatively ("negative tone") and analytically ("analytic") about health ("health") as a result of thinking about the experience of the pandemic (e.g., "illness" and "article", perhaps the result of noting "*the* pandemic"). Other PCs may hint at pre-pandemic priming leading to thinking enthusiastically about social life (PC4 –"fulfill," "prosocial," "exclamation," "social behavior"), expressing sadness over remembering one's lifestyle from the past (PC 10 –"memory," sad emotion," "leisure," "lifestyle,"), or perhaps more episodically inspired thoughts guiding future projections (PC13 –"mental," "visual," "perception," "past focus"). The during pandemic priming may lead to frustration (PC10 –"risk" and "swear") and one's needs and their justifications (PC13 –"Cause," inclusive of words like how, because, and why, and "Need"). Importantly, participants were *not* prompted to contrast the future and the present/past; the temporal prompt simply alters the semantic patterns in their writing, revealed by the PCs shown in Table 1.

One way to quantify these overall linguistic trends is to assess them using network analysis [30–32]. This method takes the PCs and visualizes the relationships among the LIWC categories. These more visual, geometric relationships among the dimensions may help to interpret the overall shift taking place in participant writing after the prompts. We built a network model using the "igraph" R package to explore which LIWC features are shared across the selected principal components (Fig 3). The nodes represent the top 50 most influential LIWC

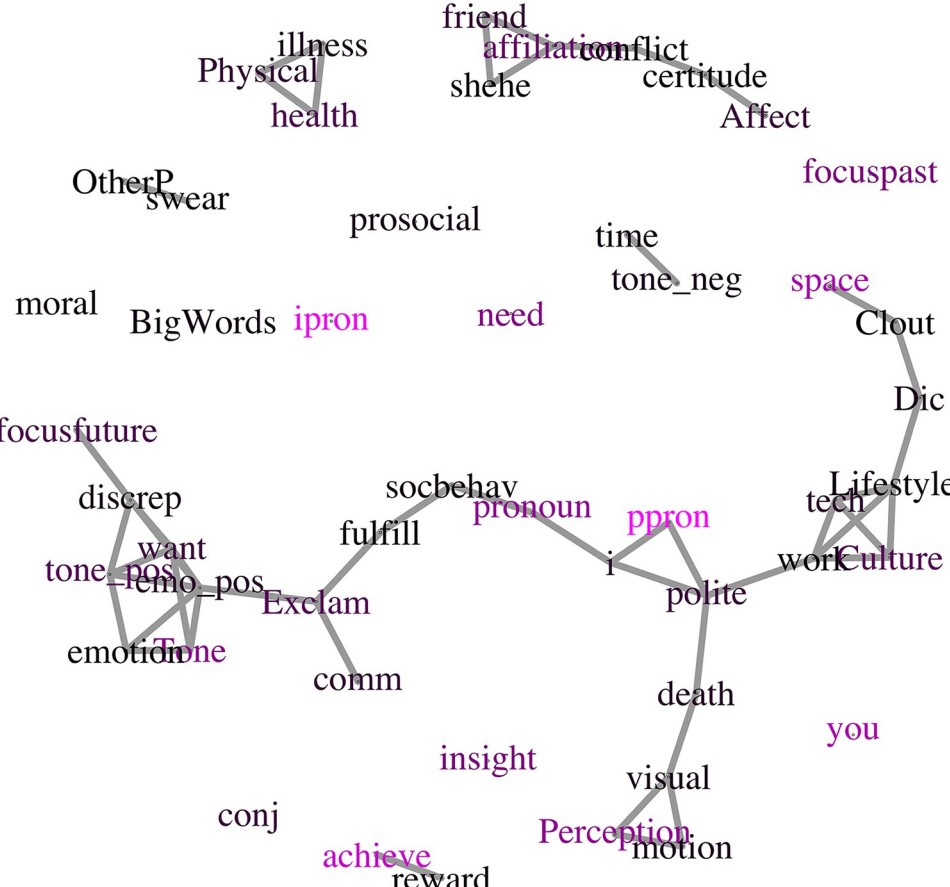

**Fig 3. Network of LIWC features shared across PCs.** LIWC features are plotted such that an edge is drawn if the features are shared across principal components 1–50. The redder the node is, the more influential the feature is in the loading scores of the main selected components (PCs 1, 4, 5, 10, 11, 13, 18).

features across the selected components (i.e., PCs 1, 4, 5, 10, 11, 13, 18). Edges are formed as the result of shared presence of the linked LIWC features across components, suggesting recurring themes in distinguishing the conditions. The color represents the level of influence that feature has in distinguishing conditions, such that the lighter the purple, the greater the influence across these components. This influence is calculated as the sum of the loading score absolute values for a given LIWC feature across components, and rather than being specific to the condition captures that feature's overall distinguishing influence.

Therefore, the lighter purple nodes represent features that are more influential across the semantic landscape. The manner in which features cluster may then represent the distinctive set of semantic factors that are combined during our particular task. For example, positive emotion, positive tone, want, tone, emotion, and discrepancy, all features found to characterize the during-pandemic condition in PC5, cluster together even across these components and appear to hint at longing for better times. Other clusters suggest livelihood elements (work, tech, lifestyle, culture), health (health, illness, physical), and sociality (she/he, friend, affiliation). When a feature within a cluster also exhibits a lighter tone, it may be the case that feature is particularly unifying of the cluster's concepts (e.g., "perception" being linked to "visual" as well as "motion"), although the feature itself may merely have more influence independently. Because semantic graphs of this kind can represent how one "moves" through meaning space (cf. [33]), a potential future application of this network-based technique is to visualize and characterize the set of potential semantic paths induced by a given prompt or frame of mind [34]. In our case, thoughts about COVID induce particular sorts of ideas, such as health and social connection. When we prompt participants with a prior temporal frame (i.e., pre-pandemic or during), they appear to take different paths on this network. Such methods may facilitate characterization of the streams-of-consciousness and internal thought processes that open this paper. Importantly though, any such graph structure should be compared to a baseline to ensure that we are not interpreting a chance outcome.

To test whether this network was structured meaningfully relative to a baseline, we extracted some measures and performed a permutation. We analyzed the mean degree (number of adjacent edges), mean betweenness (number of shortest paths going through a vertex or edge), and clustering coefficient (probability that adjacent vertices of a vertex are also connected) of the network. The mean degree is 1.92, mean betweenness 29.02, and transitivity 0.49. We then ran the same network analyses on 10,000 random permutations of the normalized LIWC data to see where the original data falls on this random distribution. This is to confirm whether such semantic structure arises specifically as a result of condition, as opposed to random clustering. The probability of the mean degree in the permutations distribution is .038,.185 for mean betweenness, and.004 for the clustering coefficient. Given both the mean degree and clustering coefficient are outside a 95% confidence interval, this suggests structure meaningfully departs from what would be expected by chance.

We next conducted a PCA on the fall (first phase) dataset. Our initial focus on the winter dataset was because we expect a more intensive response from participants–students who just had another disruption to their class activities during more lock down. To match the winter dataset, we ran a generalized linear model on the first 20 components, which account for 66% of the cumulative variance. Of these 20 components, only PC4 is significant (p = .008). The top ten loading scores for LIWC features characterizing PC4 are word count (-0.205), positive tone (0.182), impersonal pronouns (-0.181), death (-0.179), conversation (-0.17), conflict (-0.169), social references (-0.168), anger (-0.161), social (-0.157), and technology (-0.156), with each associated with the past-primed condition with the exception of positive tone. Because there is only one significant PC, we did not conduct network analysis on this dataset.

In general, there are some small effects in the fall dataset but far less pronounced than the structure we find in the winter. We return to this below.

## Conclusions and discussion

These exploratory analyses showcase the influence of temporal framing on college students as they envision their post-pandemic lives. Based only on what participants wrote when imagining their post-pandemic lives, LIWC features reduced into principal components can predict which temporal framing participants received. A few significant components hint at different categories of words that aid in making these distinctions. A tentative interpretation suggests that there is an extra focus on health (PC1) and a longing for better times (PC5) when primed by pandemic life, and more socially-oriented thinking (PC4) when primed by pre-pandemic life. In a network analysis, a semantic structure appears to arise, particularly in comparison to a distribution of random permutations of the original LIWC data. Interpretive assessment of the semantic network confirms the results on individual components. This visualization also reinforces the idea that the temporal framing leading into a stream-of-consciousness might shape the conceptual structures that participants work with. These explorations offer an initial foundation for understanding the influence of temporal thinking, and in this particular study on construing imagined futures after a major global crisis. While interpretations of the components are speculative, the LIWC categories may inform deeper studies into the specific ways COVID-19 has shaped the content of students' imagined futures. Regardless of the meaning behind these semantic spaces, this work highlights that shifts in life triggered by COVID-19 can have an impact on immediate thoughts about the future. With this in mind, interventions may be developed to explore how re-framing thoughts, such as temporally, can encourage shifts where future thoughts may be more hopeful and positive, and less dire or pessimistic.

When conducting PCA on the comparison fall dataset, we find only one component is significant. This could suggest that the winter dataset induced more complex semantics due to the emotional experiences associated with the constricted context, when students returned to remote learning due to the pandemic. Indeed this was our expectation, and motivated our initial focus on that winter data set. However, there are several factors that limit any strong conclusions. First, such environmental contextual influences need to be studied in more depth in future work. These represent just two time points, and a wider sample of data from multiple timepoints may suggest these differences were due to noise. Second, the datasets do have slightly different properties. While the winter dataset had a limited sample size, constraining the total components in the PCA, the fall dataset had a larger sample size, enabling the number of components to match the number of LIWC variables. Because of limited prior work using such methods, we did not have strong priors for an optimal sample size, which may additionally limit power in these analyses. Given our analyses were exploratory in nature, future work will benefit from taking insights gained here to formulate a priori hypotheses and planned analyses.

Priming participants to write in a stream-of-consciousness style seems less common in the literature in favor of journal paradigms, where more editing and refining of language may limit inferences about inner psychological and emotional processes, perhaps especially their dynamics. Nevertheless there are some important limitations to our own design that should be acknowledged. One limitation of the data collection using this paradigm was the online context of the study, which may include extraneous variables that would be valuable to measure and control for in future work. For example, this could include aspects of their state in the moment (e.g., exhaustion, mood), ease of technology use, and the environment when completing the task (such as presence of others in the room). Participants may also have still

performed some editing, or struggled to understand or adhere to a free-flowing style of writing. To overcome these issues, it may be useful to integrate content analysis like this with typing dynamics (e.g., [35]). Indeed, we collected individual keypresses and timings, including the use of the delete key. It may thus be possible to reconstruct some deleted content, and give a full portrait of the stream-of-consciousness exercise. These typing data may also be used to validate and refine this paradigm to study finer-grained psychological events.

Dynamic typing data may also reveal memory search, rumination through recurrent themes or word sequences. These data may also reveal document-wide typing rates that signal cognitive signatures that relate to global features such as overall sentiment and mood. In addition to word categories such as the LIWC dictionaries, other natural language processing techniques and analyses may reveal further insights. LIWC-22 also includes a measure called "narrative arc." Narrative arc includes proposed stages of composition such as staging, plot progression, and cognitive tension, and appears to follow different patterns depending on text or transcription formats, such as fictional-style writing versus *New York Times* science articles [36]. Whether journalistic or stream-of-consciousness writing follows certain narrative arc patterns, or varies depending on the topic, sentiment, or other features, remains an open question. Topic modeling or recurrence analyses can explore how possibilities become constrained (or not) by temporal framing [37, 38]. Further explorations into associations across LIWC categories could also contribute to understanding meaningful differences caused by temporal framings.

Individual differences in the experience of COVID-19 life would seem to be a critical ingredient here that we do not yet explore. Future analyses may consider such differences in more detail, such as comparing the framing texts to the post-pandemic texts. If an individual writes particularly optimistically about their life *during* the pandemic, they may be more likely to then write positively about the future, while greater negativity may similarly bleed into perceptions of the future. A rumination scale [18], a need-for-cognition scale [19], and a social connectedness and belonging scale [20] were included in data collection, though these measures were not factored into the present exploratory analyses. First-person singular pronoun use is increased in the self-focused attention typical of rumination [39], and thus may have potential for predicting rumination levels based on streams-of-consciousness. The interplay between language and rumination may result in, for example, pervasive use of such pronouns in future-oriented texts regardless of temporal framing. The need-for-cognition scale may also predict how much semantic space one covered in their streams-of-consciousness to begin with, and may inform language-oriented interventions if one temporal framing or the other encouraged greater cognitive exploration. Aside from the social connectedness and belonging measure, we asked questions about actual social experience during COVID-19. Taken together, these measures may explain some of the semantic space that the PCA revealed (e.g., PC4, which included the LIWC categories of "prosocial" and "social behavior" characterizing the pre-pandemic condition).

Given that this dataset was a college sample, factors such as age or other demographics remain open for study as they relate to global crises. For example, experiences of age-related change appear to influence perceptions of the future, and in turn mental health [40]. Age also appears to be a factor in influencing in-the-moment perceptions of COVID-19, although it may not have had as much influence on perceptions of the future [41]. While the pandemic marked a sudden major lifestyle shift globally, it will be valuable to evaluate similarities and differences to other health concerns experienced personally, such as chronic health issues, injury, or a terminal illness diagnosis. Whether the global, collective experience of COVID-19, or concern about one's own experiences drive differences in future projections remains an open question. Understanding the relationships between personal health, global health, and

how perceptions of their impact on the future change across the lifespan may clarify how different kinds of interventions may perform better or worse for different health profiles.

This study examines the effects of temporal framing on perception of the future, all within individuals; however, undoubtedly many external factors will also shape how individuals have experienced the COVID-19 pandemic. The media landscape and how the pandemic has been framed to different audiences will certainly have some influence. Indeed, psychological distance has been found to influence the evolution of misinformation regarding COVID-19 when the threats appear more distant [42]. Future work may examine the effects of the media on self-talk relating to global crises. Additionally, COVID-19 panned out to be a highly politically polarizing event. Political affiliation and intensity of an individual's antagonistic views will shape how this global event influences thoughts of a post-pandemic life. Social media usage and the makeup of one's social network both on- and off-line likely moderate perceptions of pandemic life. The language of social media posts across different styles of platforms and through different media (e.g., written such as in a tweet versus video, such as on TikTok) may show differences in socially oriented communication, which may then turn inwards when engaging in self-talk. How inter- and intra-personal communication are linked appears to be a ripe area for research [7, 43].

Our findings suggest that streams-of-consciousness could have rich dynamic properties. In broader terms, the contexts of a person's present thoughts offer a kind of momentum, propelling them into the next stream-of-consciousness. In the language of dynamical systems, there is *hysteresis*, when "the subject remains longer in the initially perceived interpretation" [44] (p. 373). This hysteresis property characterizes psychological dwell time in many domains, from motor control to categorization [45]. Even high levels of cognitive complexity, like streams-of-consciousness, may be given to these properties of complex, dynamic systems. The results here suggest this hysteresis occurs in temporal framings. When participants engage in thought about the past vs. the present, it may set their mind on a given trajectory, giving it some momentum and remaining longer in the initial perceived interpretation. Importantly though, the effects of such framings are confounded with the psychological boundary of a major global event. How context and psychological distance in the temporal domain interact or differently influence hysteresis could be the subject of future work. The global nature of COVID-19 may provide a valuable comparison point to other world events or disasters, such as the findings from Cohn, Mehl, and Pennebaker [15] of language shifts after the September 11, 2001 attacks on the United States. Future stream-of-consciousness prompts may also explore open-ended thoughts rather than anchoring on a specific event, to further clarify the temporal element alone, instead manipulating how far back or forward in time one is projecting their thoughts.

In the particular study presented here, we took an exploratory step into how streams-of-consciousness may color our views as we look towards the future. Our conscious thoughts, once expressed, are not divorced from measurable effects of thoughts that came before. The words we write and the things we think of before considering the future have a non-trivial influence on the way we frame that future to ourselves, and this can be seen even when considering a global crisis that has shaken our worlds as we knew them.

## Supporting information

**S1 Table. Loading scores for the top ten most influential LIWC features for each selected component.** The table below features the seven selected components, the ten most influential LIWC features (ordered from most to least influential, based on their absolute values), and their raw loading scores. Positive loading scores best characterize the during-pandemic

condition, while negative loading scores best characterize the pre-pandemic condition. (DOCX)

## Acknowledgments

We thank Greg Bryant and Anne Warlaumont for feedback on the project and writing, as well as the participants for contributing their streams-of-consciousness.

## Author Contributions

**Conceptualization:** Constance M. Bainbridge, Rick Dale.

**Data curation:** Constance M. Bainbridge.

**Formal analysis:** Constance M. Bainbridge, Rick Dale.

**Investigation:** Constance M. Bainbridge, Rick Dale.

**Methodology:** Constance M. Bainbridge, Rick Dale.

**Project administration:** Constance M. Bainbridge, Rick Dale.

**Resources:** Constance M. Bainbridge, Rick Dale.

**Supervision:** Rick Dale.

**Visualization:** Constance M. Bainbridge, Rick Dale.

**Writing – original draft:** Constance M. Bainbridge.

**Writing – review & editing:** Constance M. Bainbridge, Rick Dale.

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
