## [Decision Letter · Decision Letter 0]

24 Jan 2023

PONE-D-22-30477

Thinking about life in COVID-19 The influence of temporal framing on streams of consciousness

PLOS ONE

Dear Dr. Bainbridge,

Thank you for submitting your manuscript to PLOS ONE. After careful consideration, we feel that it has merit but does not fully meet PLOS ONE’s publication criteria as it currently stands. Therefore, we invite you to submit a revised version of the manuscript that addresses the points raised during the review process.

We look forward to receiving your revised manuscript.

Kind regards,

Steve Zimmerman, PhD

Associate Editor, PLOS ONE

Journal Requirements:

2. Please provide additional details regarding ethical approval in the body of your manuscript. In the Methods section, please ensure that you have specified the name of the IRB/ethics committee that approved your study.

3. Please amend your current ethics statement to address the following concerns:

a) Did participants provide their written or verbal informed consent to participate in this study?

4. Please ensure that you include a title page within your main document. You should list all authors and all affiliations as per our author instructions and clearly indicate the corresponding author.

Additional Editor Comments:

The manuscript has been evaluated by three reviewers, and their comments are available below.

The reviewers have raised a number of concerns that need attention. They request additional information on methodological aspects of the study and they request revisions to the discussion section. Could you please revise the manuscript to carefully address the concerns raised?

Reviewers' comments:

Reviewer's Responses to Questions

**Comments to the Author**

1. Is the manuscript technically sound, and do the data support the conclusions?

Reviewer #1: Partly

Reviewer #2: Yes

Reviewer #3: Yes

2. Has the statistical analysis been performed appropriately and rigorously? 

Reviewer #1: Yes

Reviewer #2: Yes

Reviewer #3: Yes

3. Have the authors made all data underlying the findings in their manuscript fully available?

Reviewer #1: Yes

Reviewer #2: Yes

Reviewer #3: Yes

4. Is the manuscript presented in an intelligible fashion and written in standard English?

Reviewer #1: No

Reviewer #2: Yes

Reviewer #3: Yes

5. Review Comments to the Author

Reviewer #1: This study appears in a qualitative form but in a quantitative form, but there are many methodological problems related to the subject of the study, which is that the subject was not studied purely empirically to ensure that the changes are due to the Corona epidemic. Including, for example, that the study was conducted online, so how can all the extraneous variables be controlled such as the ease of students’ use of technology, as well as students’ specialization, and the degree of their fatigue or exhaustion. Therefore, despite the importance of the topic, the results will be difficult to verify since the experimental steps were not followed

So I suggest putting the word exploratory study in the title

Reviewer #2: Introduction: well-written. It explains the rationale very well.

Data collection: well described

Results section:

Table 1: I would suggest to group the features by condition (i.e., colour) to make it easier to discern patterns

line 235: fraught is an odd choice of words here, what do you mean by this? Do you mean that interpreting the LIWC loadings may be subjective and thus risky? Please clarify a little.

line 235–236: what do you mean by qualitative assessment? Do you simply refer to the interpretation? If yes, the why suddenly using this term for it? It makes it seem like you mean something else than interpretation. Please make it clear what you mean (interpretation or something else)

line 237–240: I keep reading the paragraph, but keep on having trouble following it. Could you formulate your thoughts in a more accessible way? Maybe shorten the sentences? You offer a complex interpretation of the components, and it could be clearer.

Figure 3: I’m not an expert in network science, and it is not quite clear to me what the plot represents. I feel that it could be improved to be better understandable. The nodes that were most influential are coloured in a brighter red. But this information is not too useful if there is no indication of the PC the word primarily belongs to. It you added this information, it would be visible which PCSs have somewhat overlapping/related linguistic features

line 313: minor typo: while should be uppercased

Discussion section:

The discussion is very broad and touches on a very different research topics and ideas. While I do appreciate the scope, I would also be interested in how you would use the insights from this exploratory work to perform a follow-up on the ideas you present in the main body of the paper. How could the self-report measures be incorporated to supplement your findings? What could they explain in the variance of written responses/resulting LIWC dimensions?

In lines 375–389, the discussion is almost too broad and strays very far from what the content of the paper. Of course, “more work could be done in understanding other communication tools such as movement, music and other creative presentations”, however the research you just presented leaves already so many open questions that it seems a little inconsequential or far-fetched to dream of research into movement expression. Instead, I would recommend more focus in the discussion on the implications of the exploration you just presented, the paragraph starting at line 390 seems fruitful in that regard, as it presents more comprehensible implications of your work here. The preceding paragraph (375–389) could in my opinion be removed.

Please view my suggestions as well-meaning. I think the paper is well-done and a valuable, thought-provoking contribution to the field and an interesting application for LIWC. With my comments I wish to offer improvements to something that, in my eyes, is already very good and I would be happy to see it in its best possible form so that other researchers can get the most from it.

Reviewer #3: In this study, the authors examined essays written by students about life after the Covid-19 pandemic in a stream of consciousness fashion. Critically, before being prompted to write about post-pandemic life, participants were prompted to think about either 1) life before the pandemic or 2) life during the pandemic. The authors then used LIWC in combination with Principal Components Analysis to examine whether the earlier before/during prompt changed how participants wrote about the future.

Overall, I think this is a really interesting study. I liked the “stream of consciousness” paradigm and the application of LIWC, and I also think understanding how people think about the future as related to the Covid pandemic is important.

I think this manuscript is already in good shape, but I did find that the interpretation of the results was a little shallow in the results section. For instance, the authors do a good job of explaining PC5, but don’t really go into any detail about the other 6 PCs that were found to be significant in the regression model. I’m wondering what these PCs say? Based on the LIWC features shown in Table 1, the PCs seem a little hard to interpret. Maybe PC13 is related to Time in some way, but the other PCs seem a little opaque. I think more discussion of these PCs and why they might be important for distinguishing between during vs. pre-pandemic framing.

Similarly, I think more interpretation of the network analysis would be useful as well. There is a lot going on with Figure 3 as it shows the clusters of LIWC categories, but then also has colors indicating the categories that distinguish between the prompts. I wasn’t really sure what to make of this in terms of the main question of the study, which is how does thinking about pandemic vs. pre-pandemic life affect thoughts about the future. The authors again mention the words related to PC5, but I’m wondering about how some of the other categories relate? For instance, “polite”, “ppron”, “pronoun” seem to be at least somewhat clustered, and are quite red. Why would during- vs. pre-pandemic framing result in differences in these categories? Also, without having a lot of knowledge about the LIWC, it is difficult to know what some of these categories even are (e.g., I’m not really sure what “conj”, “Dic”, or “Clout” mean).

Overall, I think this is an interesting and well-written paper, and I believe that some deeper interpretations of the results would improve the manuscript and make it more interesting to readers.

6. PLOS authors have the option to publish the peer review history of their article (what does this mean?). If published, this will include your full peer review and any attached files.

Reviewer #1: **Yes: **Hanaa Shuwiekh

Reviewer #2: **Yes: **Bettina Manuela Johanna Kern

Reviewer #3: No

---

## [Author Response · Author response to Decision Letter 0]

10 Mar 2023

Dear Dr. Zimmerman,

Thank you for inviting us to revise and resubmit our manuscript, formerly entitled “Thinking about life in COVID-19 The influence of temporal framing on streams of consciousness.” The three reviewers shared some valuable criticism and we’ve addressed all their points directly in changes to the manuscript. We carefully detail our updates below with page numbers for convenience. In summary, our changes mainly included the following:

1. Highlighted the exploratory nature of our work in the title and various passages, especially in response to comments from Reviewers #1 and #2.

2. Various clarifications to the table, figures and main text in response to helpful suggestions from Reviewers #2 and #3.

3. Expanded theoretical discussion, especially in the conclusion, guided by comments from Reviewers #2 and #3.

We feel that our manuscript is much improved thanks to your input. Thank you very much to you and reviewers, and we look forward to further feedback on our resubmission.

Sincerely,

Constance Bainbridge and Rick Dale

Reviewer #1: 

This study appears in a qualitative form but in a quantitative form, but there are many methodological problems related to the subject of the study, which is that the subject was not studied purely empirically to ensure that the changes are due to the Corona epidemic. Including, for example, that the study was conducted online, so how can all the extraneous variables be controlled such as the ease of students’ use of technology, as well as students’ specialization, and the degree of their fatigue or exhaustion. Therefore, despite the importance of the topic, the results will be difficult to verify since the experimental steps were not followed

So I suggest putting the word exploratory study in the title.

Response: We have followed the reviewer’s advice and added new remarks regarding these issues (see pg. 18, lines 433-436), and we’ve also added “exploratory study” to the title.

Reviewer #2: 

Introduction: well-written. It explains the rationale very well. Data collection: well described

Response: Thank you for these encouraging remarks!

Results section:

Table 1: I would suggest to group the features by condition (i.e., colour) to make it easier to discern patterns

Response: We’ve updated the table as suggested (see pg. 12). A value discrepancy in Table 1 was also updated (the p-value for PC4, fixed from 0.037 to the correct value of 0.027).

line 235: fraught is an odd choice of words here, what do you mean by this? Do you mean that interpreting the LIWC loadings may be subjective and thus risky? Please clarify a little.

Response: We now clarify this. We agree that “fraught” may be too dire a descriptor. So we’ve reframed this passage to simply convey that any interpretations should be considered as tentative proposals for what these loadings may mean (pg. 13, line 275).

line 235–236: what do you mean by qualitative assessment? Do you simply refer to the interpretation? If yes, the why suddenly using this term for it? It makes it seem like you mean something else than interpretation. Please make it clear what you mean (interpretation or something else)

Response: This is a very helpful suggestion. We’ve updated the wording in this passage and elsewhere from “qualitative” to “interpretative,” and we agree that “qualitative” could be misinterpreted as referring to a particular family of research methods. (pg. 12, line 275; pg. 16, lines 389, 393).

line 237–240: I keep reading the paragraph, but keep on having trouble following it. Could you formulate your thoughts in a more accessible way? Maybe shorten the sentences? You offer a complex interpretation of the components, and it could be clearer.

Response: We’ve now revised this paragraph substantially to clarify our interpretation of PC5 (pg. 13, lines 276-281).

Figure 3: I’m not an expert in network science, and it is not quite clear to me what the plot represents. I feel that it could be improved to be better understandable. The nodes that were most influential are coloured in a brighter red. But this information is not too useful if there is no indication of the PC the word primarily belongs to. It you added this information, it would be visible which PCSs have somewhat overlapping/related linguistic features

Response: We’ve made a few changes to clarify this. First, we’ve expanded the accompanying description in the main text (pg. 14-15, lines 309-336), including the addition of a couple references (pg. 25, lines 695-696; 703-704). We’ve also updated the network model to provide maximally relevant information, with updated colors to make clear that it represents a different aspect of the loading scores than the condition-specific characterization.

line 313: minor typo: while should be uppercased

Response: Fixed.

Discussion section:

The discussion is very broad and touches on a very different research topics and ideas. While I do appreciate the scope, I would also be interested in how you would use the insights from this exploratory work to perform a follow-up on the ideas you present in the main body of the paper. How could the self-report measures be incorporated to supplement your findings? What could they explain in the variance of written responses/resulting LIWC dimensions?

Response: We thank the reviewers for this great suggestion. We’ve now expanded this section of our discussion. These revisit some of the key ideas of the study and how we might integrate new measures and conduct future work on related themes across COVID, health, and other domains (pg. 17, lines 404-411; pg. 19, lines 469-480 with a new reference, pg. 26, lines 734-737; pg. 20, lines 491-498; pg. 20, lines 555-560).

In lines 375–389, the discussion is almost too broad and strays very far from what the content of the paper. Of course, “more work could be done in understanding other communication tools such as movement, music and other creative presentations”, however the research you just presented leaves already so many open questions that it seems a little inconsequential or far-fetched to dream of research into movement expression. Instead, I would recommend more focus in the discussion on the implications of the exploration you just presented, the paragraph starting at line 390 seems fruitful in that regard, as it presents more comprehensible implications of your work here. The preceding paragraph (375–389) could in my opinion be removed.

Response: We’ve taken the reviewer’s helpful advice and limited the scope of the discussion to ideas more topically relevant, removing the noted paragraph (pg. 20). We are excited about the potential for this general experimental paradigm and related theoretical framework, but we agree that readers may feel jolted by the jump from our focus on pandemic-related matters into these broader topics. Because we also expanded the other discussion (see prior remark), we hope this strikes a better balance. We thank the reviewers for this suggestion.

Please view my suggestions as well-meaning. I think the paper is well-done and a valuable, thought-provoking contribution to the field and an interesting application for LIWC. With my comments I wish to offer improvements to something that, in my eyes, is already very good and I would be happy to see it in its best possible form so that other researchers can get the most from it.

Response: We greatly appreciate this constructive and helpful feedback! We hope our revisions improved the paper, and that you feel we’ve heeded your thoughtful challenges in these updates, especially in matters of methodological clarification, future directions and theoretical discussion.

Reviewer #3: 

In this study, the authors examined essays written by students about life after the Covid-19 pandemic in a stream of consciousness fashion. Critically, before being prompted to write about post-pandemic life, participants were prompted to think about either 1) life before the pandemic or 2) life during the pandemic. The authors then used LIWC in combination with Principal Components Analysis to examine whether the earlier before/during prompt changed how participants wrote about the future.

Response: Thank you for this summary, it is helpful to see how reviewers translate the main findings into a core summary. We also hope we’ve addressed your suggestions faithfully, as we summarize below.

Overall, I think this is a really interesting study. I liked the “stream of consciousness” paradigm and the application of LIWC, and I also think understanding how people think about the future as related to the Covid pandemic is important.

Response: Thank you for these encouraging remarks, we are also enthusiastic about this framing for open-ended tasks of this sort. So we’ve kept some of these broader points in both the introduction and conclusion, and also highlighted specific issues in response to Reviewer #2 and your comments below.

I think this manuscript is already in good shape, but I did find that the interpretation of the results was a little shallow in the results section. For instance, the authors do a good job of explaining PC5, but don’t really go into any detail about the other 6 PCs that were found to be significant in the regression model. I’m wondering what these PCs say? Based on the LIWC features shown in Table 1, the PCs seem a little hard to interpret. Maybe PC13 is related to Time in some way, but the other PCs seem a little opaque. I think more discussion of these PCs and why they might be important for distinguishing between during vs. pre-pandemic framing.

Response: This is helpful, and like Reviewer #2 we think Reviewer #3 highlights some needed expansion of our discussion. This expansion is framed around the particular study and its results, and we now elaborate a bit more about the PCs (pg. 13, lines 282-291) and also, as suggested by Reviewer #2 as well, add clarification on the implications of the specific study in this domain in the general discussion (pg. 17, lines 404-411; pg. 19, lines 469-480).

Similarly, I think more interpretation of the network analysis would be useful as well. There is a lot going on with Figure 3 as it shows the clusters of LIWC categories, but then also has colors indicating the categories that distinguish between the prompts. I wasn’t really sure what to make of this in terms of the main question of the study, which is how does thinking about pandemic vs. pre-pandemic life affect thoughts about the future. The authors again mention the words related to PC5, but I’m wondering about how some of the other categories relate? For instance, “polite”, “ppron”, “pronoun” seem to be at least somewhat clustered, and are quite red. Why would during- vs. pre-pandemic framing result in differences in these categories? Also, without having a lot of knowledge about the LIWC, it is difficult to know what some of these categories even are (e.g., I’m not really sure what “conj”, “Dic”, or “Clout” mean).

Response: It is very helpful to get these remarks, and indeed Reviewer #2 shares similar suggestions (see points above). We mentioned above and we note again here for convenience: We’ve now expanded our discussion of the network diagram and updated the diagram to provide more meaningful information via color, shifting to a different color to avoid confusion with condition-related color choices in the Table and Fig. 2. We’ve also elaborated more on the interpretation of the network (pg. 14-15, lines 309-336). Additionally, we note a reference to the LIWC-22 manual (Boyd et al., 2022), which includes further details on what the LIWC categories encapsulate (pg. 11, lines 231-232).

Overall, I think this is an interesting and well-written paper, and I believe that some deeper interpretations of the results would improve the manuscript and make it more interesting to readers.

Response: Thank you! We hope our revisions have achieved this important direction, and we greatly appreciate your feedback.

---

## [Decision Letter · Decision Letter 1]

3 Apr 2023

PONE-D-22-30477R1Thinking about life in COVID-19: An exploratory study on the influence of temporal framing on streams-of-consciousnessPLOS ONE

Dear Dr. Bainbridge,

Thank you for submitting your manuscript to PLOS ONE. After careful consideration, we feel that it has merit but does not fully meet PLOS ONE’s publication criteria as it currently stands. Therefore, we invite you to submit a revised version of the manuscript that addresses the points raised during the review process.

We look forward to receiving your revised manuscript.

Kind regards,

Michal Ptaszynski, PhD

Academic Editor

PLOS ONE

Journal Requirements:

Reviewers' comments:

Reviewer's Responses to Questions

**Comments to the Author**

1. If the authors have adequately addressed your comments raised in a previous round of review and you feel that this manuscript is now acceptable for publication, you may indicate that here to bypass the “Comments to the Author” section, enter your conflict of interest statement in the “Confidential to Editor” section, and submit your "Accept" recommendation.

Reviewer #1: All comments have been addressed

Reviewer #2: All comments have been addressed

Reviewer #3: All comments have been addressed

2. Is the manuscript technically sound, and do the data support the conclusions?

Reviewer #1: Partly

Reviewer #2: Yes

Reviewer #3: Yes

3. Has the statistical analysis been performed appropriately and rigorously? 

Reviewer #1: Yes

Reviewer #2: Yes

Reviewer #3: Yes

4. Have the authors made all data underlying the findings in their manuscript fully available?

Reviewer #1: Yes

Reviewer #2: Yes

Reviewer #3: Yes

5. Is the manuscript presented in an intelligible fashion and written in standard English?

Reviewer #1: Yes

Reviewer #2: Yes

Reviewer #3: Yes

6. Review Comments to the Author

Reviewer #1: The study seems quantitative, but it was dealt with qualitatively, and therefore the results are more like suggestions and not facts, which confirms that if the application is repeated again on these students, will the same results be obtained? I expect not.

Therefore, I suggest adding the word exploratory study in the title of the study

Reviewer #2: Dear authors, I'm very impressed by your work, congratulations! Thank you for considering my suggestions and for your kind remarks on them. I hope to see your paper published soon.

All the best!

Reviewer #3: The authors have addressed all of my comments in the previous round, and I believe the manuscript, which was already in good shape, is even stronger now. In particular, I found the additional paragraphs on interpreting the PCs and network analysis were very helpful. Overall, I think this will make a very nice contribution to PLOS ONE, and I appreciate the opportunity to review this work!

7. PLOS authors have the option to publish the peer review history of their article (what does this mean?). If published, this will include your full peer review and any attached files.

Reviewer #1: **Yes: **Hanaa Shuwiekh

Reviewer #2: **Yes: **Bettina MJ Kern

Reviewer #3: No

---

## [Author Response · Author response to Decision Letter 1]

7 Apr 2023

Reviewer #1: 

The study seems quantitative, but it was dealt with qualitatively, and therefore the results are more like suggestions and not facts, which confirms that if the application is repeated again on these students, will the same results be obtained? I expect not.

Therefore, I suggest adding the word exploratory study in the title of the study

Response: We hope to clarify this point: we had updated the title of the manuscript to include “exploratory study” in our previous revision, honoring this very helpful comment from Reviewer #1. We noticed the “short title” we submitted into the system still used the former title, so we have additionally updated the short title to now read “Thinking about life in COVID-19: An exploratory study on the influence of temporal framing.” If there is an alternate spot in the manuscript this comment intended to address, we are happy to make any additional updates to make sure this aspect of the research is clear. We have resubmitted again and double checked that “exploratory” is indeed in the title of the manuscript. If this is not visible to reviewers or editor, we welcome any advice about correcting it. Thank you again for your suggestion!

Reviewer #2: Dear authors, I'm very impressed by your work, congratulations! Thank you for considering my suggestions and for your kind remarks on them. I hope to see your paper published soon.

All the best!

Reviewer #3: The authors have addressed all of my comments in the previous round, and I believe the manuscript, which was already in good shape, is even stronger now. In particular, I found the additional paragraphs on interpreting the PCs and network analysis were very helpful. Overall, I think this will make a very nice contribution to PLOS ONE, and I appreciate the opportunity to review this work!

Response to all reviews: Thank you once again for your valuable feedback in strengthening the manuscript!

---

## [Decision Letter · Decision Letter 2]

18 Apr 2023

Thinking about life in COVID-19: An exploratory study on the influence of temporal framing on streams-of-consciousness

PONE-D-22-30477R2

Dear Dr. Bainbridge,

We’re pleased to inform you that your manuscript has been judged scientifically suitable for publication and will be formally accepted for publication once it meets all outstanding technical requirements.

Kind regards,

Michal Ptaszynski, PhD

Academic Editor

PLOS ONE

Additional Editor Comments (optional):

Reviewers' comments:

Reviewer's Responses to Questions

**Comments to the Author**

1. If the authors have adequately addressed your comments raised in a previous round of review and you feel that this manuscript is now acceptable for publication, you may indicate that here to bypass the “Comments to the Author” section, enter your conflict of interest statement in the “Confidential to Editor” section, and submit your "Accept" recommendation.

Reviewer #1: (No Response)

2. Is the manuscript technically sound, and do the data support the conclusions?

Reviewer #1: (No Response)

3. Has the statistical analysis been performed appropriately and rigorously? 

Reviewer #1: (No Response)

4. Have the authors made all data underlying the findings in their manuscript fully available?

Reviewer #1: (No Response)

5. Is the manuscript presented in an intelligible fashion and written in standard English?

Reviewer #1: (No Response)

6. Review Comments to the Author

Reviewer #1: Thank you for accept my suggestion and adding "“exploratory study”" " in the title, Thank you again

7. PLOS authors have the option to publish the peer review history of their article (what does this mean?). If published, this will include your full peer review and any attached files.

Reviewer #1: **Yes: **hanaa Shuwiekh

---

## [Editor Report · Acceptance letter]

20 Apr 2023

PONE-D-22-30477R2 

Thinking about life in COVID-19:
An exploratory study on the influence of temporal framing on streams-of-consciousness 

Dear Dr. Bainbridge:

I'm pleased to inform you that your manuscript has been deemed suitable for publication in PLOS ONE. Congratulations! Your manuscript is now with our production department. 

Kind regards, 

on behalf of

Dr. Michal Ptaszynski 

Academic Editor

PLOS ONE